# Design of Miniaturized Wideband Beam Deflection Conformal Array Antenna

**DOI:** 10.3390/s23020930

**Published:** 2023-01-13

**Authors:** Junli Zhu, Chuanyong Sun, Mengfei Chen, Jingping Liu

**Affiliations:** School of Electronic and Optical Engineering, Nanjing University of Science and Technology, Xiaolingwei Street, Nanjing 210094, China

**Keywords:** microstrip antenna, beam deflection, cylindrical conformal, artificial electromagnetic structure

## Abstract

Antenna beam deflection, along with miniaturization and wideband of the antenna is in demand for practical applications. In this paper, a cylindrical conformal array antenna with a small-tilt forward beam was designed. The microstrip antenna unit was loaded with the artificial electromagnetic structure, which reduced the size of the antenna unit. As a result, the center spacing of the array elements can be shortened with the same array element spacing. The beam deflection angle can be increased in this way without increasing the coupling effect between the parts. Changing the number of line array elements and the number of line arrays can regulate the beam width of E-field and H-field, respectively. The bandwidth of the antenna can be significantly extended by slotting the ground plane. This work implemented a cylindrical conformal array of the antenna’s forward beam with a small dip angle using a cylindrical carrier as an example. The measurement results showed that the angle between the main beam and the carrier axis of the conformal antenna was less than 30°, the bandwidth was more than 30%, and the antenna volume decreased by 40.4%.

## 1. Introduction

Low frequencies require antennas with large physical sizes, which hinders the miniaturization and integration development of wireless communication systems. There are various microstrip antenna miniaturization technologies, but antenna size reduction deteriorates antenna performance. For example, when antenna size is reduced, the gain of the antenna decreases correspondingly, and antenna bandwidth decreases. The literature [1] proposes to change the current distribution on the surface of the antenna by etching the patch surface to make the effective current path longer and reduce the resonant frequency to reduce the size. Although the size is reduced by 17%, bandwidth and gain are also reduced. In the literature [2], antenna size can be reduced by using a high impedance surface photonic crystal structure, and performance parameters can be maintained. Ceramic materials have been widely used in radio frequency, ultra-wideband, mobile phones, and other wireless communication systems in recent years [3]. Their high dielectric constant helps to realize the miniaturization of the antenna and reduce the bandwidth of the antenna. In this paper, the artificial electromagnetic structure was etched on the microstrip patch, which reduced the area of the microstrip antenna unit by 44.7%.

When reducing the antenna size, antenna performance must also be considered to minimize the possibility of other performance degradations. The bandwidth of the antenna is an important performance parameter related to working performance. The greatest feature of the microstrip antenna used in this paper is that it is very thin, which makes it conformal with the flying body. However, the main disadvantage of the microstrip antenna is the narrow frequency band, as the antenna is required to have a wide frequency band in many applications. The narrowband characteristic of the microstrip patch antenna limits its application in microwave field. To solve the problem of antenna miniaturization and wideband, researchers have proposed various methods, such as: (1) using structural symmetry: Sun M and Zhang Y P [4] symmetrically selected the half-edge structure of a planar monopole antenna made by a low-temperature co-fired ceramic technology, so that the antenna structure was reduced by 40%. The miniaturized bevelled monopole antenna measuring 17 × 10 × 1 mm achieved an impedance bandwidth of 8.25 GHz from 2.85 to 11.1 GHz; and (2) slot for patch or ground plate: Majidzadeh M [5] et al. designed a quad radiation patch with steps and inserted a rectangular slot and three steps in the ground plane, thus obtaining new resonance and a wider bandwidth. The antenna had a compact size of 18 × l2 × l.6 mnl and covered a frequency range of 2.34–21.43 GHz (9.15:1; 160%). [6] presented a printed monopole antenna with stable omnidirectional radiation patterns. The frequency selective surface is combined with the cross-ground structure to expand the bandwidth from 1.7–2 GHz to 1.65–4 GHz. In this paper, slots are made in the ground plane of the microstrip antenna, so that the bandwidth of the antenna is extended from 5.75–5.88 GHz to 5.26–6.63 GHz.

In addition to the antenna volume and bandwidth, the directivity of the antenna is strictly required in some practical applications. An independent antenna unit usually cannot control the beam. It usually requires a microstrip array antenna or loading special structures on the antenna unit to control the beam width, deflection angle, and sidelobe level of the antenna. For example, the Antenna Technology Branch of the AIR Force Sensor Agency developed digital beamforming (DBF) technology for beam control [7]. Reference [8] controlled the deflection angle of the beam of the antenna by connecting a matching load at the end of the microstrip serial-fed antenna array. Ref. [9] showed that the meta-surface antenna can provide beam steering by phasing all the resonators appropriately. In the literature [10], omnidirectional radiation was achieved by controlling the number of antenna units distributed on the surface of the conformation to control the irradiation range of the antenna beam. Sheng Lili et al. [11] proposed a beam deflection and wide/narrow beam switching antenna based on a monolayer digital meta-surface. The beam can be controlled by controlling the biasing voltage of PIN diodes in each column. The beam deflection range is 0° to ±45° at the central operating frequency of 3.6 GHz. Ming Li et al. [12] designed a 1 × 16 linear irregular frequency reconfigurable antenna array (FRAA) with beam deflection ability based on the proposed FRA element. However, the antenna structure is complex, the feed network occupies a large volume, and the maximum beam deflection angle is only 20°. As the beam deflection angle of a series-fed array is influenced by the center distance of array elements, the beam deflection of a large angle needs a small array element spacing, and the small distance between array elements produces coupling, which negatively affects the antenna radiation. Therefore, it is difficult to achieve large angle beam deflection. In this paper, we used the method of loading artificial electromagnetic structures to reduce the volume of microstrip antenna array elements, so that the center distance of array elements is reduced with the same distance between adjacent array elements, thus realizing a larger angle beam deflection. The antenna designed in this paper achieves a beam deflection angle of 60°.

The cylindrical conformal antenna composed of a microstrip antenna with beam deflection and a cylindrical carrier can be widely used in the aerospace field. The conformal antenna integrates with the carrier. Early conformal antennas were usually applied to the surfaces of non-planar carriers. The advantages of aconforming antenna and carrier are: it does not destroy the mechanical structure of the carrier; it does not produce additional aerodynamic resistance; and it has little impact on the aerodynamic performance of the carrier itself. At the same time, it saves space and reduces the radar reflection area. These advantages can be used in optimization to reduce the aircraft’s radar characteristics [7]. Ref. [13] designed an X-band hemispheric space beam covered satellite mobile communication antenna conformal with the surface of an elliptical cylinder. Han Dongbo and Hong Fei [14] designed an UWB (ultra-wideband) special microstrip antenna with a warhead type based on the microstrip antenna. However, the conformal antenna was located inside the carrier, which offset the ground profile characteristics of the patch antenna and occupied a large space. T P S Kumar Kusumanchi et al. [15] produced a novel approach for the simulation of hexagonal conformal array (HCA) antenna design. [16] proposed an extremely ultrathin 2-D isotropic flexible meta-surface energy harvester (MEH) using a multi-polarized patch frequency selective surface. This MEH, which has a curved shape, enjoys a relatively constant performance over a huge range of oblique incident waves. The final structure can be prepared by a small slice of a cylinder. The design in [17,18] designed a new flexible ultra-thin curvature meta-surface energy harvester. The design paves the way for the multi-polarized MEH to wrap around the cylindrical surface as a 2D isotropic MEH. 

With all the advantages of a conformal antenna and microstrip antenna, the microstrip conformal antenna has become one of the research hotspots of antenna technology and has been widely used in radar, communication, and other fields. The cylindrical aircraft needs to perceive the cone-shaped space at a certain angle with its axis direction during flight. The beam formed by the deflected microstrip antenna and the cylinder conformal to the cylinder has a certain angle with the axis of the cylinder. A MIMO omnidirectional conformal array composed of beam-deflected microstrip linear arrays, when used in an active array technology and can be electronically steered. Along with the rotation of the aircraft, beam scanning in a conical space can be realized. This provides a new idea for the cone beamforming of cylindrical conformal antennas.

In this paper, a series-fed microstrip antenna array with beam deflection was designed, and the artificial electromagnetic structure was loaded to improve the controllability of beam deflection and antenna miniaturization. The antenna bandwidth was extended by slotting the ground plate, and the microstrip line array of beam deflection was formed into a plane array, which meets the design requirements of H-field beam width. In addition, the planar antenna was conformed with the cylinder, and the miniaturized wideband conformal array of the beam deflection was designed.

## 2. Beam Deflection of Series-Fed Microstrip Linear Array

The network structure of the series-fed array is simple and compact, and the connected feeder is relatively short. The beam deflection of the main lobe of the array radiation pattern can adjust the spacing of the radiation elements of the array antenna, and the bandwidth of the antenna is widened.

The model of the microstrip array antenna in the form of a serial feed is shown in Figure 1, with one end of the linear array feeding. The phase relation between the elements can be changed by adjusting the distance between the array elements, which deflects the beam direction arbitrarily, and shifts the main beam.

Assuming that the angle between the main beam of the antenna and the *z*-axis is *θ*, then the relationship between the main beam pointing angle of the antenna array and the spacing of the antenna array element is [19]:(1)$$\cos\theta =\lambda /\lambda_{g}-\lambda /S$$

*S* refers to the spacing between array elements. That is, *S* is the sum of the distance *d* between adjacent array elements and the length *L* of the non-radiative edge of array elements, *λ* is the wavelength in air, and *λ_g_* is the effective wavelength in the medium. This formula is an approximation and can be used as a reference for design, without considering the impact of discontinuity at the joint. The spacing of radiation elements *S* is one of the most important parameters affecting the radiation characteristics of an antenna array. When the resonant frequency of the antenna is determined and the spacing between the array elements *S* is equal to the effective wavelength in the medium, the main beam of the microstrip series-fed array is directed at 90° to the *z*-axis, that is, the *y*-axis direction, and the beam does not deflect. If *S* is reduced by *λ_g_*, the antenna beam is deflected to the *z*-axis.

Assuming the center frequency is 5.8 GHz of the designed antenna, using Rogres5880 as the substrate with a relative dielectric constant *ε_r_* of 2.2, the angle between the main beam and the shaft is less than 30°. In this paper, the electromagnetic simulation software Ansoft HFSS was used to simulate the antenna model.

According to Formula (1), a reference value of distance *d* between array elements is 8.2 mm. The array element is not an ideal power source in practical design, as radiation, reflection, and phase shift of the array element need to be considered. The array element spacing *d* obtained by simulating the electromagnetic simulation environment with HFSS software will change relative to the reference value. Simulation results showed that when the distance *d* between adjacent array elements is 4.5 mm, the angle between the antenna beam pointing and the positive *z*-axis is 30°, that is, the antenna beam is deflected by 60° from the positive *y*-axis to the positive *z*-axis. The simulated distance *d* between adjacent array elements is less than the calculated value, which reduces the distance between adjacent array elements and strengthens the electromagnetic coupling, thus affecting the radiation of the antenna. Therefore, there are certain limitations in the method of controlling beam deflection by simply changing the spacing of array elements.

For the broadside array, the beam width of the main lobe is [20].
(2)$${\left(BW\right)}_{bh}=0.886\lambda /\left(NS\right)\left(rad\right)=51\left(\lambda /E\right)\left({}^{\circ}\right)$$

*E = NS* is the length of the array, *N* is the number of array elements, and *d* is the distance between any adjacent array elements. According to Equation (2), the beam width of the broadside array is inversely proportional to the length of the array. Element spacing *d* affects the main beam deflection of the microstrip linear array, and the influence of element number of a linear array on the beam width is shown in Table 1.

According to the simulation results in Table 1, the beam width of the E-field of the microstrip linear array decreases as the number of array elements increases, i.e., the angle of the E-field decreases, which is consistent with Formula (2). As the increase in the number of elements mainly affects the pattern of the E-field, the influence on the pattern of the H-field is minor, which is shown in Table 1 that the beam width of the H-field does not fluctuate much.

Considering the requirement on 20° beam width of the index to the E-field diagram, eight radiation units are selected for the microstrip feeder array.

The array antenna is simulated and optimized with HFSS software. By analyzing the parameter of the size of the adversary element and the spacing of the array element, the size of the element is 19.5 mm × 17 mm, and the spacing of the element is *d* = 4.5 mm. The simulation results of S11 of 1 × 8 microstrip serial-fed array antenna are shown in Figure 2. The simulated bandwidth of the antenna is 5.75–5.88 GHz, and the impedance bandwidth is 2.23%. Compared with the bandwidth of a single rectangular microstrip patch antenna, the bandwidth of the array antenna is widened. Simulation results of 1 × 8 microstrip serial-fed array antenna pattern of E-field and H-field at 5.8 GHz are shown in Figure 3. The angle between the main beam of E-field and the *z*-axis is 30°. The maximum gain of the antenna is 12.18 dB, the beam width of the E-field is 19.95°, and the beam width of the H-field is 83.4° at 5.8 GHz.

## 3. Analysis of the Influence of Artificial Electromagnetic Structure on Beam Deflection

The artificial electromagnetic structure of the I-shaped resonant ring is loaded on the rectangular microstrip antenna to reduce the resonant frequency of the antenna, and then by reducing the physical size of the antenna radiating patch, the resonant frequency is increased back to the original center frequency of 5.8 GHz, thereby realizing miniaturization of the antenna patches.

In this paper, the I-shaped resonant ring was selected as the artificial electromagnetic structure to be etched on the microstrip antenna. Reference [21] used the transmission line theory to analyze the principle of the patch for etching the artificial electromagnetic structure. Figure 4 shows the I-shaped resonant ring model, in which *a* is the side length of the resonant ring, *w*1 is the longitudinal microstrip line width of the resonant ring, and *w*2 is the horizontal microstrip line width of the resonant ring. The dimensions of the I-shaped resonator and the design method of the patch loaded with the structure have been described in detail by the authors in [22].

First, the resonant frequency of the microstrip antenna unit under different sizes of the I-shaped resonant ring is simulated to determine the size of the I-shaped resonant ring. Then, different numbers of resonant rings are etched on the microstrip unit, and the influence of the number of resonant rings on the S parameters of the microstrip patch unit is analyzed by simulation. Finally, the size of the patch unit is changed so that the antenna resonates at 5.8 GHz.

The microstrip antenna model of etched I-shaped resonant ring is shown in Figure 5. The change of the resonance frequency with the side length *a* of the I-shaped resonance ring is shown in Figure 6. When the side length *a* increases, the resonance frequency decreases continuously. From the miniaturization effect of the antenna, the larger the side length *a* of the resonant ring, the better. However, considering the overall size of the microstrip patch in actual design, *a* = 2 mm is selected as the side length of the etched I-shaped resonant ring.

To analyze the influence of the number of artificial electromagnetic structures on the miniaturization effect of the antenna, the simulation analysis of the rectangular microstrip antenna with different numbers of I-shaped resonant rings etched by HFSS software is carried out. Figure 7 is a diagram showing the relationship between the resonant frequency of the microstrip antenna and the number of etched I-shaped resonant rings. Etching the I-shaped resonant ring on the microstrip antenna reduces the resonant frequency of the antenna. As the number of resonant rings increases, the resonant frequency decreases gradually.

Finally, the resonant frequency of the antenna is returned to 5.8 GHz by reducing the length of the rectangular microstrip antenna to realize the miniaturization of the antenna, and at the same time adjusting the width of the patch antenna to complete the impedance matching of the miniaturized antenna.

After the simulation analysis of the size of the I-shaped resonant ring, we determined the side length, the vertical microstrip line width and the horizontal microstrip line width of the resonant ring etched on the microstrip patch antenna, which are respectively *a* = 2 mm, *w*1 = 0.5 mm, *w*2 = 0.2 mm. The miniaturized microstrip antenna model with 15 I-shaped resonator rings etched on the rectangular microstrip antenna is shown in Figure 5.

After realizing impedance matching, the size of the 5.8 GHz miniaturized microstrip antenna is *W* = 15 mm and *L* = 13.2 mm. The size of the rectangular microstrip patch designed according to the transmission line theory is *W* = 21 mm and *L* = 17.05 mm. The area of the microstrip antenna patch was reduced by 44.7%.

The beam is only able to be deflected in a limited range by spacing control but loading the artificial electromagnetic structure can increase the deflection angle by reducing the length *L* of the non-radiating side of the patch without changing the distance *d* between adjacent array elements. Its influence on the main beam deflection of the antenna pattern is analyzed below.

One row, two rows, and three rows of evenly spaced I-shaped resonance rings are arranged on the unit patch of the microstrip serial feed antenna array, and the model diagram of the antenna is shown in Figure 8.

HFSS electromagnetic simulation software is used for analysis, and the simulation results of the deflection angle of the antenna pattern are shown in Figure 9.

The size of the antenna element of the I-shaped resonant ring array etched in different numbers and the angular deflection of the main beam in the pattern of the E-field are compared, and the results are shown in Table 2.

Keeping the same reference calculation value of spacing *d* = 8.2 mm between array antenna elements, Figure 9 shows that the angle between the main beam of the array antenna and the *z*-axis decreases gradually as the number of I-shaped resonance rings increases. Table 2 shows the size of the unit patch of the corresponding array. The etching of the artificial electromagnetic structure reduces the size of the array significantly. The main beam of the antenna is reduced from the initial angle of 48° to 30° with the *z*-axis, that is, the deflection angle is increased from 42° to 60°.

The 1 × 8 microstrip serial-fed array antenna model shown in Figure 8c was optimized. Its simulation results of S11 is shown in Figure 10. The relative bandwidth of the array antenna is about 3%. The simulation result of the pattern of the array antenna at 5.8 GHz is shown in Figure 11. The optimized spacing between antennas is *d* = 6.1 mm, the maximum gain of antennas is 11.7 dB, and the angle between the main beam of the E-field and the *z*-axis is 29°. The beam width of the E-field is 23.4°. The width of the beam of the H-field is 64.5°. The overall size of the array antenna is reduced by 40.4% compared with that of the microstrip array antenna without miniaturization, which indicates the significance of the miniaturization.

## 4. Broadband Plane Array and Conformal Array Antenna

The narrow-band characteristics of microstrip antennas limit their wide use. To overcome this obstacle and meet the demand for the wide-band of microstrip antenna, this paper explores a design method for the wide-band of microstrip antenna. Slots are made for the ground plate part at the back of each array element to broaden its bandwidth. It is assumed that the center frequency of the designed antenna is 5.8 GHz, and the relative bandwidth is no less than 15%. 

As shown in Figure 12, the microstrip antenna units of the planar array are all arranged in the xoz plane. *N*_1_ rows in the *x*-axis direction, the spacing of the elements is *S*_1_; *N*_2_ columns in the *z*-axis direction, and the array element spacing is *S*_2_. The microstrip antenna element located in row *m* and column *n* is denoted as element *mn*.

For a uniformly distributed planar lateral array, when *N*_1_ and *N*_2_ are large, the width [20] of the main lobe in the two main planes are
(3)$$\begin{array}{l} {\left(BW\right)}_{xoz}=51\frac{\lambda }{N_{1}S_{1}}\left({}^{\circ}\right)\\ {\left(BW\right)}_{yoz}=51\frac{\lambda }{N_{2}S_{2}}\left({}^{\circ}\right) \end{array}$$

By analyzing the beam width of the linear array, the requirement can be satisfied when the number of elements in the *z*-axis direction is 8. Suppose that the beam width of the E-field of the designed antenna is 20°. According to the formula, when *N*_1_ = 2, the width of the H-field is 50°. The modle of 2 × 8 miniaturized microstrip serial feed array antenna slotted in ground plate is shown in Figure 13.

HFSS software was used for parameter analysis, and the optimized S11 parameters are shown in Figure 14. The reflection coefficient of the antenna at 5.8 GHz is −24.7 dB, and the relative bandwidth of the antenna is 23.0%. Compared with the 3% bandwidth of the miniaturized planar array antenna, the bandwidth is expanded significantly. The disadvantage of the antenna’s narrow operating band is overcome, which increases the application range of the series-fed microstrip antenna. In the bandwidth of the antenna, the maximum gain of the antenna is relatively stable.

The pattern of the planar array antenna is shown in Figure 15. Once the ground plate is slotted, the antenna produces back radiation, and part of the energy is lost. The maximum gain of the antenna is 11.46 dB. The deflection angle of the beam of the E-field is 29°. And its half-power beam width is 21.8°, and the width of the half-power beam width of the H-field is 49.7° at 5.8 GHz. After applying the slotting technology to the ground plate, although the gain of the antenna decreases from 11.7 dB to 11.46 dB at 5.8 GHz, the beam deflection direction of the antenna remains unchanged. And the maximum gain of 1 × 8 wideband miniaturized microstrip serial-fed array antenna over bandwidth is shown in Figure 16.

A conformal antenna is an antenna with the same shape as the carrier on which the antenna is placed. The conformal antenna needs to be attached to the surface of the carrier, and it requires a simple structure, high reliability, compact sizing, and lightweight. The low profile of the microstrip conformal antenna lets it conform well with various surfaces, which leads to wide use and a great impact on the development of related fields.

A cylinder with a diameter of 50 mm is used as the antenna carrier, and a flexible dielectric substrate with Rogres5880 is adopted. Its relative dielectric constant ε_r_ is 2.2, and the thickness *h* = 0.254 mm. 

The antenna model diagram of the miniaturized broadband 2 × 8 conformal array in HFSS simulation software is shown in Figure 17.

With consideration of the one-to-two power divider and the parameters of the array element, the antenna parameters are analyzed and simulated. The optimized antenna parameters are as Figure 18 The simulation shows that the abscissa at the lowest point of reflection coefficient −30 dB is 5.8 GHz. The antenna return loss is small, the relative bandwidth of the antenna ranges from 5.38 GHz to 6.89 GHz, and the bandwidth is about 24%. After simulation optimization, the size of the conformal array antenna element is 14.47 mm × 10.72 mm. According to Figure 19, the angle between the main beam and the *z*-axis is 30° at 5.8 GHz, the half-power beam width is 21.5°, and the gain of the antenna is 9.2 dB. The main beam is nearby without deflection, and the beam width of the simulation is 50.49°. For comparison, Table 3 shows the model data and simulation results of the 4 different antenna models designed in this paper.

Figure 20 shows the front and back of the miniaturized wideband 2 × 8 conformal array antenna when it is tiled. Figure 21 shows the antenna conformed to a cylinder with a diameter of 50 mm.

Figure 18 shows the comparison between measured and simulation results of S parameters of 2 × 8 conformal array antenna. The resonance point of the measured convolution antenna array is 5.78 GHz, which shifts to the left slightly compared to the simulation result. The S_11_ is −26 dB at the operating frequency of 5.8 GHz, and the return loss is small. The absolute bandwidth of the measured antenna ranges from 4.74 GHz to 6.76 GHz. Its relative bandwidth is about 34.8%, which is wider than that of the simulation, and it meets the design index of bandwidth >15%. Simulation and testing show that ground plane slotting widens the antenna bandwidth when the antenna is mounted on a metal object. Due to the insufficient precision of some thinner feeders during the antenna processing, or the influence of the amount of solder between the feeder and the joint, and the connection between the adapters during measurement, some errors are introduced in the measurement. The measured S parameter of the 2 × 8 conformal array antenna performs well, and the relative bandwidth meets the requirements of the design index. The measured and simulated S-parameter trends are consistent.

The radiation characteristic parameters of the conformal antenna were measured in the microwave anechoic chamber. According to the comparison between the measured and simulated normalized pattern of E-field at 5.8 GHz, the trends of the pattern are mostly consistent. The abscissa of the maximum measured gain direction is 29.3°, which meets the design requirements. The half-power beam angle of the E-field ranges from 20° to 39°, that is, the beam width is 19°, which is slightly narrower than the simulated beam width. 

From the E-field pattern of the antenna, it can be seen that the antenna has a high-level lobe in the -yoz plane, that is, in the direction of the back of the cylinder, which is mainly due to the diffraction of the electromagnetic wave near the patch around the cylinder. The superposition of electromagnetic waves diffracted by different elements forms the beam on the back of the cylinder. When the operating frequency of the microstrip antenna increases, the wavelength becomes shorter, and the backward beam level caused by diffraction will decrease. Alternatively, increasing the diameter of the cylinder will also reduce the back beam of the conformal antenna, while keeping the operating frequency and physical size of the antenna the same. The design of this paper aims to provide a cone scanning beam for a high-speed rotating cylindrical aircraft. The angle between the beam and the cylinder axis is less than 30°, which is within the design requirements. Adding a lobe on the back of the cylinder can double the scanning speed when the roll angular velocity of the aircraft remains unchanged. The ability has been greatly improved. 

The beam of the measured pattern of H-field moves slightly to the right, and the beam width of the H-field is 50.9°. In the actual measurement, there may be some measurement errors due to deflection angle control. The measurement of the radiation pattern is mostly consistent with the simulation results. According to the method of antenna gain measurement, antenna gain size is 7.6 dB.

## 5. Conclusions

This paper introduced the design process of a miniaturized wideband beam deflection conformal array antenna. First, the effects of array spacing and artificial electromagnetic structures loaded on beam deflection direction were discussed. The beam direction of E-field was deflected by adjusting the length of feeders between microstrip liner array elements. It was demonstrated that loading artificial electromagnetic structures can change the direction of the current in the patch, reduce the patch length, further reduce the array spacing, increase the degree of beam deflection, and achieve an angle of 30° with *z*-axis. The relationship between the number of line elements and the beam widths of E-field, the number of line arrays, and the beam widths of H-field were discussed, respectively. It was shown that when the number of line elements and line arrays increases, the beam widths of E-field and H-field decrease, respectively. In this paper, the line array adopted eight elements and the line array consisted of two line arrays. The antenna bandwidth was increased from 3% to 23% by slotting the ground floor to reduce the Q value of the antenna. Finally, combined with the design method of cylindrical conformal antenna, the plane array was conformed with the cylinder. Two antennas were designed with an angle of 30° between the E-field beam and the *z*-axis, and H-field beam width of 84° and 50°, respectively. The two antennas realized the formation of the forward beam of the cylindrical conformal array antenna with a small tilt angle.

## Figures and Tables

**Figure 1 sensors-23-00930-f001:**
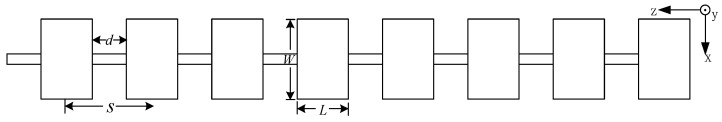
Series-fed microstrip patch array.

**Figure 2 sensors-23-00930-f002:**
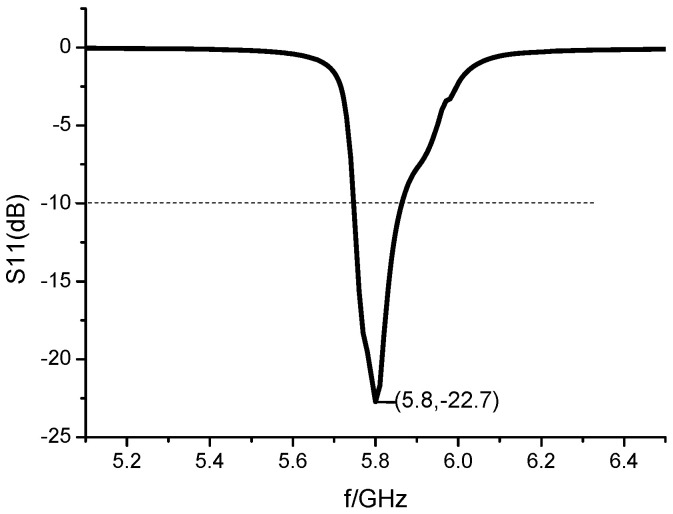
The simulation results of S11 of 1 × 8 microstrip serial-fed array antenna.

**Figure 3 sensors-23-00930-f003:**
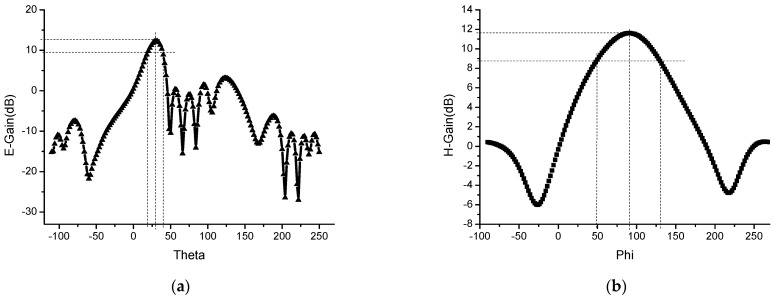
Simulation results of 1 × 8 microstrip serial-fed array antenna pattern of E-field and H-field at 5.8 GHz: (**a**) E-field; and (**b**) H-field.

**Figure 4 sensors-23-00930-f004:**
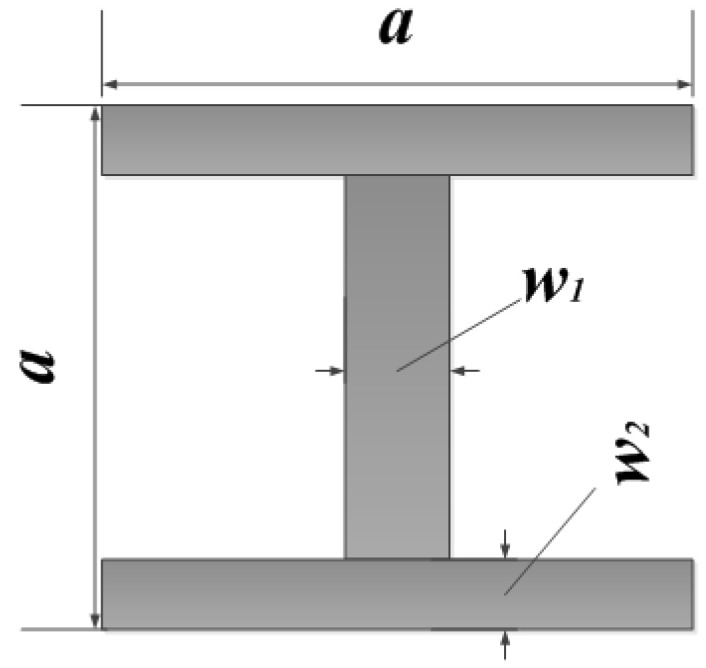
The structure of the I-shaped resonant ring.

**Figure 5 sensors-23-00930-f005:**
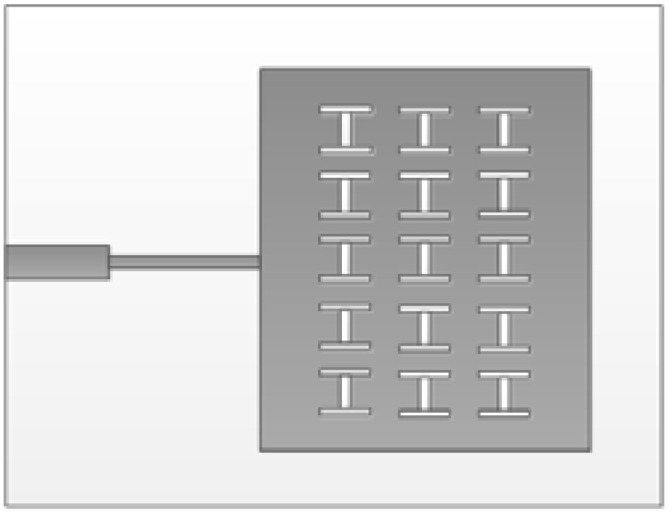
Microstrip antenna model of etched I-shaped resonant ring.

**Figure 6 sensors-23-00930-f006:**
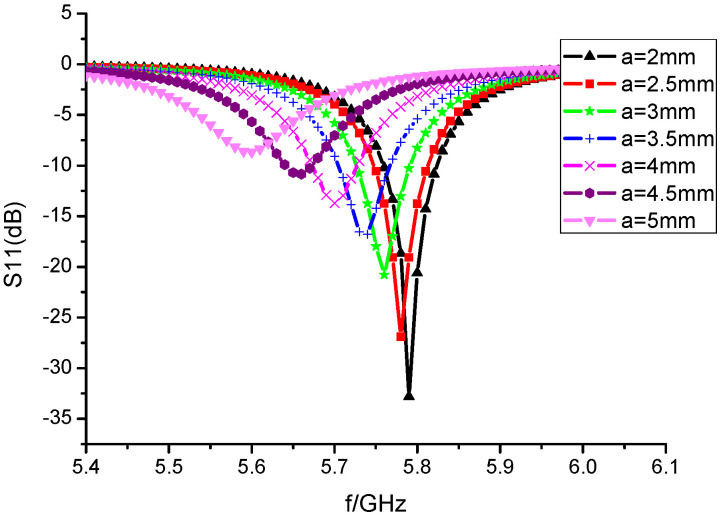
Variation diagram of resonance frequency with side length *a*.

**Figure 7 sensors-23-00930-f007:**
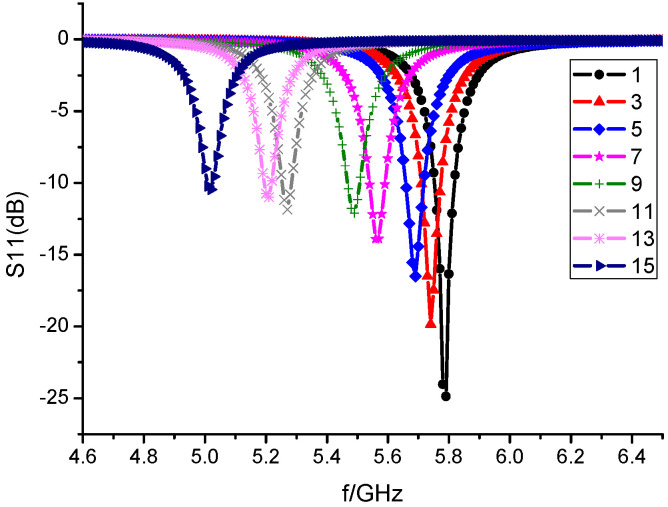
Variation of resonance frequency with the number of resonant rings.

**Figure 8 sensors-23-00930-f008:**
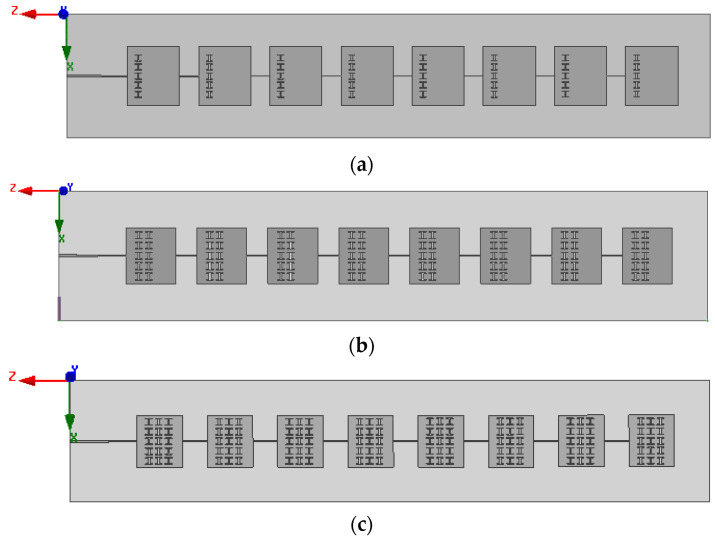
1 × 8 microstrip serial feed array antenna etched with artificial electromagnetic structure: (**a**) unit etches a row of I-shaped resonance rings; (**b**) unit etches two rows of I-shaped resonance rings; and (**c**) unit etches three rows of I-shaped resonance rings.

**Figure 9 sensors-23-00930-f009:**
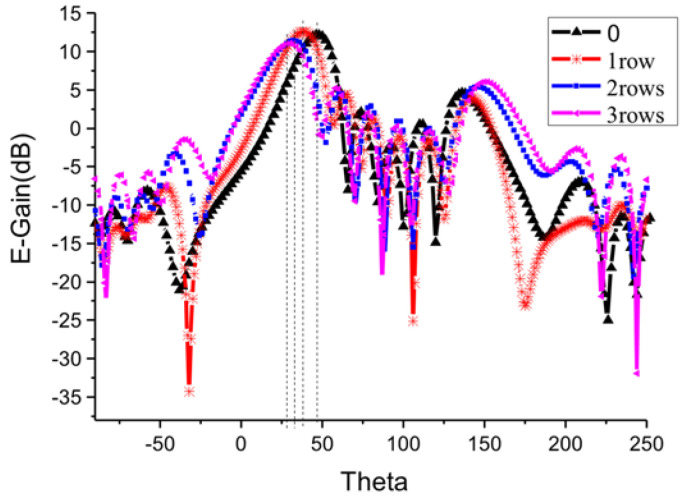
Influence of etched artificial electromagnetic structure on main beam deflection of antenna E-field pattern.

**Figure 10 sensors-23-00930-f010:**
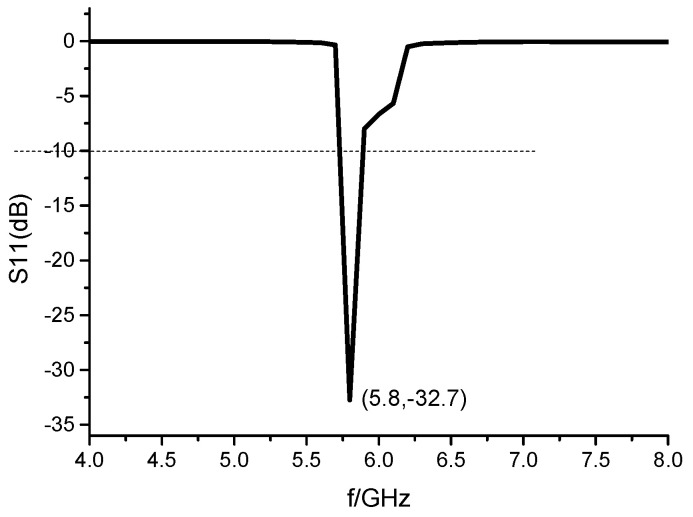
Simulation results of S_11_ of 1 × 8 miniaturized microstrip serial fed array antenna.

**Figure 11 sensors-23-00930-f011:**
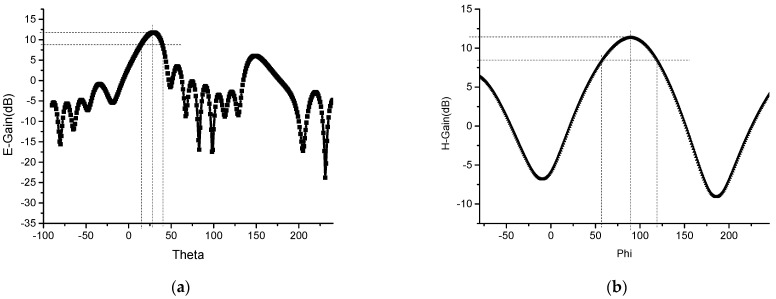
Simulation results of 1 × 8 miniaturized microstrip serial-fed array antenna pattern of E-field and H-field at 5.8 GHz: (**a**) E-field; and (**b**) H-field.

**Figure 12 sensors-23-00930-f012:**
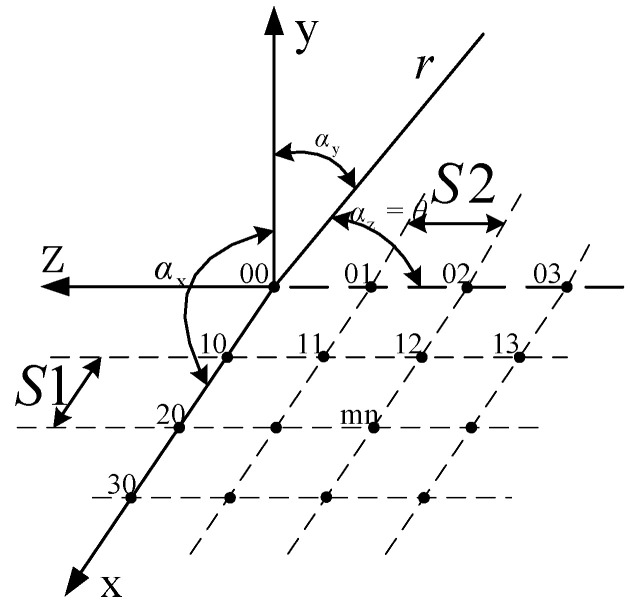
Space diagram of planar array.

**Figure 13 sensors-23-00930-f013:**
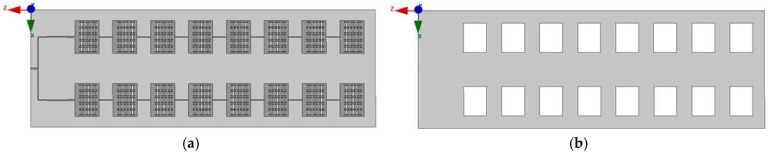
2 × 8 miniaturized microstrip serial feed array antenna slotted in ground plate: (**a**) front; and (**b**) back.

**Figure 14 sensors-23-00930-f014:**
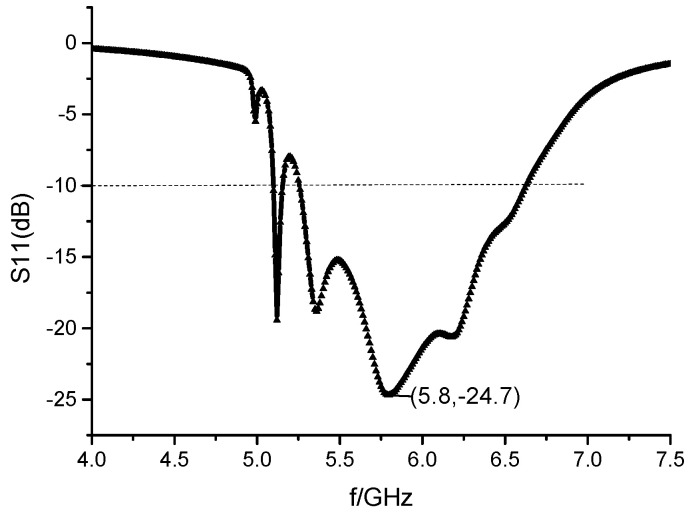
The simulation results of S11 of miniaturized 2 × 8 planar array antenna of the slotted ground plate.

**Figure 15 sensors-23-00930-f015:**
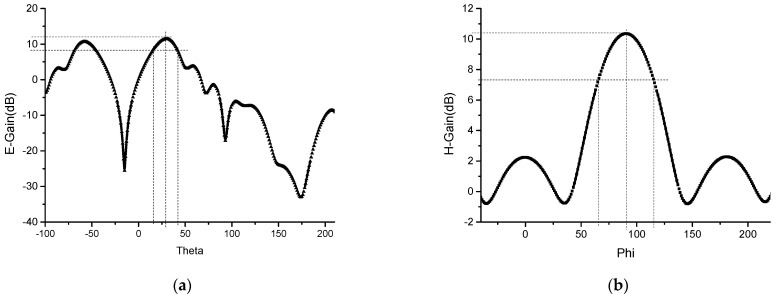
Simulation results of miniaturized 2 × 8 planar array antenna pattern for slotted ground plate at 5.8 GHz: (**a**) E-field; and (**b**) H-field.

**Figure 16 sensors-23-00930-f016:**
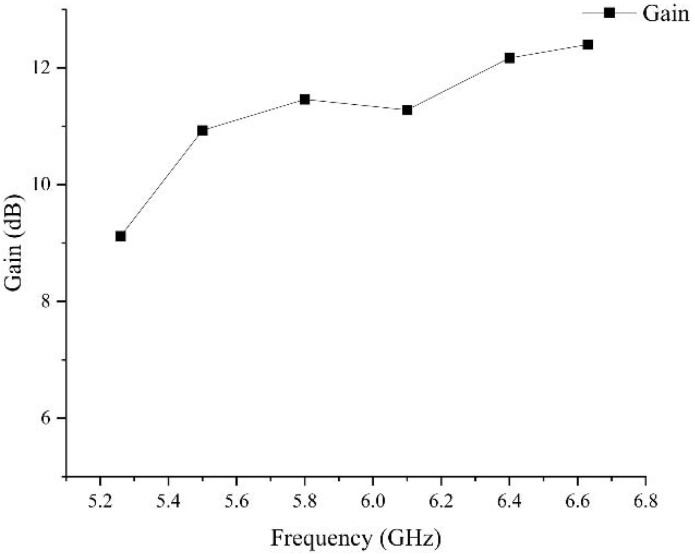
The maximum gain of 1 × 8 wideband miniaturized microstrip serial-fed array antenna over bandwidth.

**Figure 17 sensors-23-00930-f017:**
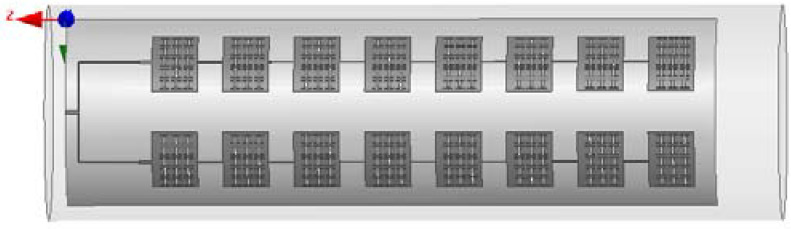
Miniaturized broadband 2 × 8 conformal array antenna model.

**Figure 18 sensors-23-00930-f018:**
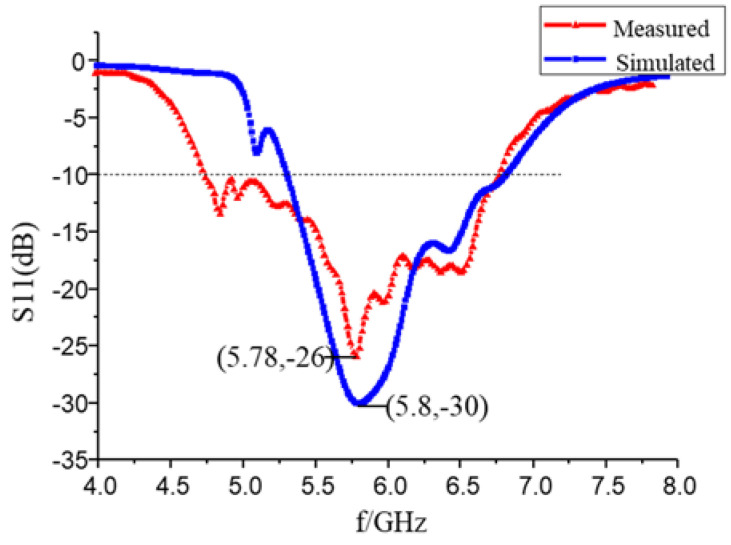
Comparison between actual measurement and simulation of 2 × 8 conformal array antenna.

**Figure 19 sensors-23-00930-f019:**
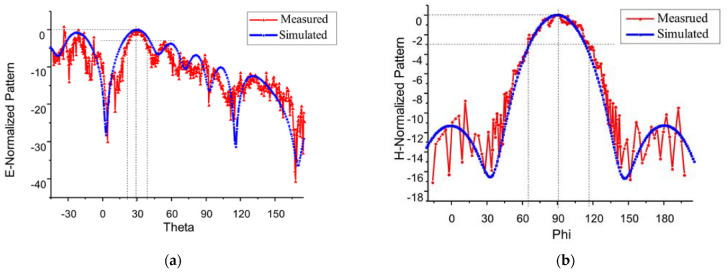
The normalized pattern of miniaturized wideband 2 × 8 conformal array antenna measured and simulated comparison at 5.8 GHz: (**a**) E-field; and (**b**) H-field.

**Figure 20 sensors-23-00930-f020:**
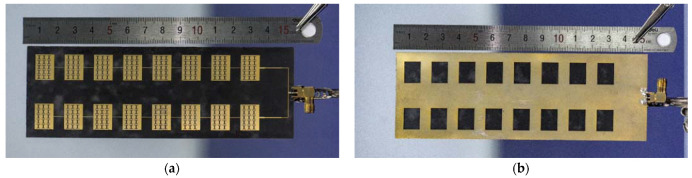
The physical picture of miniaturized wideband 2 × 8 conformal array antenna: (**a**) front; and (**b**) back.

**Figure 21 sensors-23-00930-f021:**
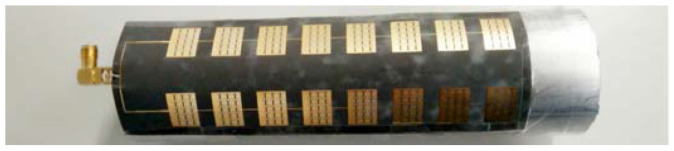
The physical picture of miniaturized wideband 2 × 8 conformal array antenna.

**Table 1 sensors-23-00930-t001:** Orientation diagram of a microstrip array antenna with different number of elements beam width.

Array Element Number/N	6	7	8	9	10	11
Main lobe width of E-field (deg)	23.9	21.7	20.96	18	17.35	16.42
Main lobe width of H-field (deg)	86.8	94.46	94.46	93.5	94.47	90.7

**Table 2 sensors-23-00930-t002:** Simulation results of microstrip array antennas etched with artificial electromagnetic structures at 5.8 GHz.

Unit Size	Unetched (0)	Single Row (5)	Two Rows (10)	Three Rows (15)
W/mm	19.5	18.3	15.1	14.9
L/mm	17	16.3	14.2	13
Angle between main beam and *z*-axis/deg	48	39	33	30
Gain/dB	12.23	12.08	11.45	11.03

**Table 3 sensors-23-00930-t003:** Simulation parameters of 4 different antenna models.

Model	1 × 8 Line Array	Miniaturized 1 × 8 Line Array	2 × 8 Wideband Planar Array	2 × 8 Cylindrical Conformal Array
Element size *W × L*	19.5 × 17 mm^2^	15.1 × 13 mm^2^	14.5 × 10.66 mm^2^	14.47 × 10.72 mm^2^
Element spacing *d*	4.5 mm	6.1 mm	6.1 mm	6.1 mm
Thickness *h*	0.254 mm	0.254 mm	0.254 mm	0.254 mm
Load artificial electromagnetic structure	No	Yes	Yes	Yes
Slot the floor	No	No	Yes	Yes
Cylindrical conformal	No	No	No	Yes
Bandwidth (GHz)	5.75–5.88 (2.23%)	5.7–5.88 (3.1%)	5.26–6.63 (23.0%)	5.38–6.89 (24%)
Gain at 5.8 GHz (dB)	12.18	11.7	11.46	9.2
Beam deflection angle at 5.8 GHz (deg)	60	61	61	60
Beam width of E-field at 5.8 GHz (deg)	19.95	23.4	21.8	21.5
Beam width of H-field at 5.8 GHz (deg)	83.4	64.5	49.7	50.49

## Data Availability

Not applicable.

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
