# Peer review of "Design of Miniaturized Wideband Beam Deflection Conformal Array Antenna"

_sensors, 2023, doi:10.3390/s23020930_

Round 1
Reviewer 1 Report
The paper deals with the deflection of the main beam by controlling the spacing of the array elements and loading the artificial electromagnetic structure. This is one of the emerging topics and is fully in the scope of the journal. The paper structure is not clear. There are a few aspects to be mentioned:
· In the abstract the sentence” In order to form the forward beamforming of the cylindrical con-8 formal array antenna with a small tilt angle, this paper explores the deflection of the main beam and 9 miniaturization of the antenna by controlling the spacing of the array elements of the microstrip 10 series-fed array antenna and loading the metamaterial artificial electromagnetic structure” is too long and not easily understandable. Please rewrite the sentence again.
· An abstract needs to be rewritten.
· No continuity in the introduction. it needs to be correct
· The authors have used the terms “metamaterial artificial electromagnetic structure” in some parts of the text and in some other parts only “artificial electromagnetic structure”. Please correct the term and use only either one of the following terms: metamaterial structure or artificial electromagnetic structure.
· What is the importance of artificial electromagnetic structures? Explain in detail by providing the simulation analysis of I- the shaped resonating ring.
· The authors may compare the proposed antenna with existing literature in terms of miniaturization and bandwidth at their respective operating frequencies.
· Please indicate the realized gain instead of the gain total.
· Please show the beam width of both the E-plane and H-plane in all cases and compare each other
· There is another resonance frequency at about 5GHz in a 2×8 planar array antenna. What is the effect of it
· Compare the S-parameter analysis in all cases
· Please compare reflection parameters graphically in all the cases to understand the enhancement of bandwidth. Also compare in a table.
· Almost 10% bandwidth difference is there between simulation and measurement. Why?
(Planar bandwidth 23%, conformal bandwidth 24%, measure bandwidth 34.8 %)
Author Response
The responses to the reviewers' comments are in the appendix. Reviewers can turn on revision mode to see the record of changes made to the author's article.

Reviewer 2 Report
The manuscript presents a design of a miniaturized wideband beam deflection conformal array antenna; however, the manuscript is difficult to follow. In addition, some shortcomings prevent me from recommending its publication. I recommend the author consider my comments provided in the following and resubmit a substantially revised paper. Here are my comments and suggestions for the authors:
1- English must be checked.
2- Please prepare a fair comparison table so that the advantages and disadvantages of the designed antenna can be demonstrated. For a fair comparison, the table should contain the size, bandwidth, gain, beam deflection, operation frequency, thickness, etc.
3- In the abstract, the authors stated that “The antenna bandwidth can be extended by slotting the ground plate”; In the conclusion, the authors stated that “Then the antenna bandwidth is increased from 3% to 23% by slotting the ground floor to reduce the Q value of the antenna”. “ the deflection angle is increased from 42° to 60°.”
Since the aim of the work is to design an antenna with three important characteristics:
1-miniaturization, 2-wideband, and 3-beam deflection, it is better to add figures that show clearly them and their application in the real world. For example, in order to show the effect of metamaterial on the deflected beams please show the simulation results of patterns with different dimensional sizes of I-shaped resonance rings. Please add figures that show the simulation results of the bandwidth of the proposed antenna ( “2×8 array antenna”) with and without slotting the ground plate.
4- In the conclusion, the authors stated that “This paper introduces the design process of a miniaturized wideband beam deflection conformal array antenna”. Please, explain how you did design the dimensions of the I-shaped resonance rings? Also, it is not clear why they chose the I-shaped resonance rings at 5.8GHz with 2mm×2mm size.
5- Please, state that the test results claimed in the abstract are from simulation or measurement results.
6- The references used in this manuscript are old. Out of thirty-three references in the manuscript, about twenty-one references are before 2012 (or fourteen references are before 2004).
7- Metasurface collectors and also cylindrical conformal metasurface arrays were analyzed and investigated by other researchers. The references are not adequate, many similar works have not been reviewed in the manuscript. For example:
[a] Analysis, Design, and Implementation of a New Extremely Ultrathin 2-D-Isotropic Flexible Energy Harvester Using Symmetric Patch FSS, doi: 10.1109/TMTT.2020.2982386
[b]An extremely ultrathin flexible Huygens’s transformer, https://doi.org/10.1063/5.0016373
[c] A True Metasurface Antenna, https://www.nature.com/articles/srep19268
[d] An ultra-thin double-functional Metasurface patch antenna for UHF RFID applications, https://www.nature.com/articles/s41598-020-79506-5
[e] A new compact dual-band perfect absorption ultrathin planar Metasurface energy harvester in X- and V-bands with a wide incident angle, https://aip.scitation.org/doi/full/10.1063/5.0012857
[f] Monopole Antenna with Enhanced Bandwidth and Stable Radiation Patterns Using Metasurface and Cross-Ground Structure, https://www.mdpi.com/1424-8220/22/21/8571
[g] Design and fabrication of a 2D-isotropic Flexible ultra-thin Metasurface for ambient electromagnetic energy harvesting, https://doi.org/10.1063/1.5083876
8 8-The reference does not meet the format requirements of MDPI journals. Please correct the format of references in the manuscript.
9- References have been used that are not easy to find online, please provide their access link so that the reader can easily refer to them for further reading. For example, five M.S. theses [9,10,13,32,33] and [12] are cited, please provide their access link in the manuscript. The doi provided for references [12], [13], [32] and [33] are not working.
110- Please indicate, for the sake of clarity, what frequency is employed in the results of Table II.
111- In line 246, the authors stated that “The etching of the artificial electromagnetic structure reduces the size of the array significantly”. Why?
Author Response

(The authors gave the same response as above.)

Round 2
Reviewer 1 Report
The authors have improved the manuscript.
In my opinion it may be accepted for publication
Reviewer 2 Report
Authors have considered most of my comments.